# The Xpert® MTB/RIF diagnostic test for pulmonary and extrapulmonary tuberculosis in immunocompetent and immunocompromised patients: Benefits and experiences over 2 years in different clinical contexts

Ana Paula de Oliveira Tomaz [1,2☯¤a], Sonia Mara Raboni[2☯], Gislene Maria Botão Kussen[2☯], Keite da Silva Nogueira[1,2‡], Clea Elisa Lopes Ribeiro[3‡], Libera Maria Dalla Costa[1,2☯¤b]*

**1** Programa de Pós graduação em Biotecnologia Aplicada à Saúde da Criança e do Adolescente da Faculdades Pequeno Príncipe (FPP), Instituto de Pesquisa Pelé Pequeno Príncipe (IPPPP), Curitiba, Paraná, Brasil, **2** Complexo Hospital de Clínicas, Universidade Federal do Paraná (CHC-UFPR), Setor de Infectologia, Setor de Bacteriologia, Unidade de Laboratório de Análises Clínicas (ULAC) Curitiba, Paraná, Brasil, **3** Secretaria Municipal da Saúde, Setor Vigilância Epidemiológica de HIV/AIDS, Curitiba, Paraná, Brasil

☯ These authors contributed equally to this work.
¤a Current address: Complexo Hospital de Clínicas, Universidade Federal do Paraná (CHC-UFPR), Curitiba, Paraná, Brasil
¤b Current address: Instituto de Pesquisa Pelé Pequeno Príncipe (IPPPP), Curitiba, Paraná, Brasil
‡ These authors also contributed equally to this work.
* lmdallacosta@gmail.com

## Abstract

Xpert® MTB/RIF has been widely used for tuberculosis (TB) diagnosis in Brazil, since 2014. This prospective observational study aimed to evaluate the performance of Xpert in different contexts during a two-year period: (i) laboratory and clinical/epidemiological diagnosis; (ii) HIV-positive and -negative populations; (iii) type of specimens: pulmonary and extrapulmonary. Overall, 924 specimens from 743 patients were evaluated. The performance of the assays was evaluated considering culture (Lowenstein Jensen or LJ medium) results and composite reference standard (CRS) classification as gold standard. According to CRS evaluation, 219 cases (29.5%) were classified as positive cases, 157 (21.1%) as 'possible TB', and 367 (49.3%) as 'not TB'. Based on culture, Xpert and AFB smear achieved a sensitivity of 96% and 62%, respectively, while based on CRS, the sensitivities of Xpert, AFB smear, and culture were 40.7%, 20%, and 25%, respectively. The pooled sensitivity and specificity of Xpert were 96% and 94%, respectively. Metric evaluations were similar between pulmonary and extrapulmonary samples against culture, whereas compared to CRS, the sensitivities were 44.6% and 29.3% for the pulmonary and extrapulmonary cases, respectively. The Xpert detected 42/69 (60.9%) patients with confirmed TB and negative culture on LJ medium, and 52/69 (75.4%) patients with negative AFB smear results. There was no significant difference in the diagnostic accuracy based on the types of specimens

**Data Availability Statement:** All relevant data are within the paper.

**Funding:** This study was partially financed by the Coordenação de Aperfeiçoamento de Pessoal de Nível Superior - Brazil (CAPES) – Financing Code 001. The funders had no role in study design, data collection and interpretation, or the decision to submit the work for publication. This funding covers only the payment of a scholarship to pay the student's tuition to the institution (FPP). This amount did not cover the cost of the project.

**Competing interests:** The authors have declared that no competing interests exist.

and population (positive- and negative-HIV). Molecular testing detected 13 cases of TB in culture-negative patients with severe immunosuppression. Resistance to rifampicin was detected in seven samples. Herein, Xpert showed improved detection of pulmonary and extrapulmonary TB cases, both among HIV-positive and -negative patients, even in cases with advanced immunosuppression, thereby performing better than multiple other diagnostic parameters.

## Introduction

Globally, tuberculosis (TB) poses one of the most significant health threats and is among the 10 leading causes of deaths from infectious diseases, besides AIDS, malaria, and currently, COVID-19 [1]. According to the Global Tuberculosis Report of World Health Organization (WHO), approximately 10 million people in 2019 were affected with TB, causing 1.4 million deaths; of these, 208,000 were HIV-positive individuals [2]. In Brazil, 73,864 cases of infection were diagnosed, with an incidence of 35 cases/100,000 habitants in 2019 [3].

The delay in diagnosis due to test limitations and the similarities of symptoms with other respiratory diseases may contribute to TB dissemination. The chances of contracting TB infection are 26 times higher in HIV-infected individuals compared with the general population. Moreover, it is associated with high morbidity and mortality in immunosuppressed patients. Thus, early diagnosis and treatment are essential for effective TB control, especially in patients who have a broad-spectrum disease with atypical, extrapulmonary, and paucibacillary cases [1,4,5].

In this context, molecular detection of *Mycobacterium tuberculosis* (MTB) has advantages in diagnosis, including speed, standardization, and high yield. Among the molecular diagnostic options available, the GeneXpert® MTB/RIF system (Cepheid Inc. Sunnyvale, CA, USA) has been approved by WHO since 2010. A revised guidelines were published in 2013 endorsing the use for diagnostic pulmonary TB, pediatric TB, extrapulmonary TB, rifampicin resistance, and for TB/HIV coinfection cases. In Brazil, this rapid molecular test (TRM) was introduced, in 2014 [6,7]. However, conventional methods of smears and cultures continue to be considered the gold standard for diagnosis and monitoring the treatment for TB, despite their limitations such as low sensitivity and long incubation period [8–11].

This study aimed to evaluate the performance of Xpert in comparison with traditional methodologies in different contexts, namely, laboratory and clinical-epidemiological diagnosis, distinct clinical samples (pulmonary and extrapulmonary), and in positive- and negative-HIV patients. A composite reference standard (CRS) and culture results in Lowenstein Jensen (LJ) medium were the gold standard methods for comparison.

## Materials and methods

### Study population and specimens

This was a prospective observational study conducted at a tertiary academic hospital of the Federal University of Paraná (CHC/UFPR) between May 2015 and June 2017. Pulmonary and extrapulmonary samples were collected from patients with suspected TB, with and without immunosuppression, from nine public institutions, including: CHC/UFPR, emergency care units (UPA), hospitals, and outpatients from Curitiba and metropolitan regions. Patients whose samples did not show valid culture results (due to contamination or insufficient

sample), had volumes less than 2 mL or had mycobacteria identified from the non-TB group (MNT) were excluded. The study was approved by the local Institutional Ethics Review Board (N# 37624214.3.0000.0096), and all participants signed the informed consent form.

## Laboratory methods

Sample transport, processing, and direct detection of MTB by acid-fast bacillus (AFB) smear staining, as well as culture (LJ) were conducted based on the manual/WHO/2015 [12]. MTB isolates were identified using TB Ag MPT64 BIOEASY identification test (Standard Diagnostic, Republic of Korea). Drug susceptibility testing (DST) was conducted using BACTEC 460 TB system (Becton Dickinson Microbiology Systems, Sparks, Md), according to WHO recommendations [13].

Molecular Xpert MTB/RIF assay was performed with a fraction of the sample using on the GeneXpert® system (Cepheid, Sunnyvale, CA, USA) in accordance with the manufacturer's instructions, and the Xpert MTB/RIF implementation manual/WHO/2014 [14,15]. Urine samples were processed as per the body fluids protocol; thus, 1 mL of the urine sample was mixed with 2 mL of Xpert sample reagent [16].

The AFB smear, LJ culture and Xpert assay test were performed at CHC/UFPR, and DST for *M. tuberculosis* was performed at the Reference Public Health Laboratory of Paraná (LACEN).

## Clinical data and TB case definitions

Confirmed cases of TB reported were identified from the National Information System of Notification Diseases (SINAN); HIV laboratory data (CD4+ and viral loads results were obtained from Laboratory Examinations Control System (SISCEL). Patients were followed up for two years after samples collection to determine possible and discarded TB cases.

The analyzed variables were classified into three groups: 1) Sociodemographic profile: Gender, age, underlying disease, and chemical dependency (illicit drugs, smoking, or alcoholism); 2) clinical and epidemiological profile: Contact with TB, type of case (new case, relapse, reentry after abandonment or transfer), clinical outcome, clinical presentation, presumptive diagnosis (imaging tests, histopathology), HIV serology, HIV viral load, current and nadir T-CD4+ lymphocyte count (cells/μL), antiretroviral treatment, deaths and their causes; 3) laboratory: results of smears, cultures, and Xpert tests.

The results of the Xpert were compared with those of the gold culture standard. However, since conventional culture (LJ) media is suboptimal in detecting paucibacillary samples, a CRS was used [16–18]. This study adopted the diagnostic criteria for active TB established by SINAN for the investigation, notification, follow-up, and treatment of communicable diseases. SINAN enables continuous data consolidation, monitoring, and evaluation of actions on TB control Nationwide, which combines laboratory and clinical-epidemiological data. After searching and analyzing the SINAN database, patients were classified, based on the CRS, into 4 groups: (i) "Confirmed TB" (AFB-positive/culture-negative, AFB-negative/culture-positive and AFB/culture-positive patients); (ii) "Probable TB" (AFB/culture-negative patients, presence of clinical symptoms of TB, radiological findings, and/or histology suggestive of TB); (iii) "Possible TB" (AFB/culture-negative patients), presence of clinical signs/symptoms of TB but without any records for the treatment of TB); (iv) "Not TB" (negative results for all tests and treatment not registered in the TB database) [19–21]. Confirmed TB patients and probable TB were classified as CRS-positive cases and Possible TB and not TB as CRS-negative cases. Table 1 lists the algorithm used for patient categorization.

**Table 1. Algorithm for patient categorization into different categories of the composite reference standard.**

| CRS category | Results | | | | | |
|---|---|---|---|---|---|---|
| | AFB smear (n = 924) | Culture (n = 924) | Symptoms/Signs [a] or contact for TB (n = 395) | Radiology [b] (n = 269) | Histology [c] (n = 36) | Follow-up [d] at 2 years at SINAN (n = 219) |
| Confirmed TB (n = 69) | +/- | +/- | + | +/- | +/- | + |
| Probable TB (n = 150) | - | - | + | + | + | + |
| | - | - | + | + | - | + |
| | - | - | + | - | + | + |
| Possible TB (n = 157) | - | - | + | - | - | - |
| Not TB (n = 367) | - | - | +/- | - | - | - |

CRS: Composite reference standard.

AFB: Acid-fast bacillus.

[a]Symptoms include coughing, sweating, fever, weight loss, tiredness, and recent HIV diagnosis. Contact with individuals, including bacilliferous subjects, at home or outside.

[b]the specimen was probable or confirmed, according to the patient's medical record or notification form.

[c]a specimen was positive when the presence of AFB is detected (or suggestive).

[d]a specimen was positive if the patient was on antitubercular treatment (ATT) based on the National Information System of Notification Diseases (SINAN) classification, and negative when no such notification is provided.

## Statistical analysis

Sociodemographic and clinical-epidemiological data were analyzed using the JMP® statistical software. The VENN diagrams were constructed using the online tool <https://bioinfogp.cnb.csic.es/tools/venny/> and the kappa coefficient was calculated to analyze the agreement between the methods and/or final diagnosis [22]. Chi-square test was used for the statistical comparison of categorical variables. A P-value < 0.05 was considered statistically significant.

We evaluated sensitivity and specificity with 95% confidence intervals; positive and negative predictive values of the molecular test were compared with the culture results and CRS classification. Forest plot graphs were generated to display the estimates of sensitivity and specificity using Microsoft Excel®. Further, the analysis of Xpert test performance was conducted by comparing the type of samples analyzed (pulmonary/extrapulmonary) and the HIV-positive/HIV-negative patients.

## Results

### Study population

In total, 743 patients were included in the study; of these, 616 patients underwent AFB smear, culture, and Xpert tests using a single sample, whereas 127 patients underwent multiple samples. Based on CRS-positive (confirmed TB/probable TB) classification, there were 219 (29.5%) cases of TB, 157 (21.1%) 'possible TB' cases, and 367 (49.3%) cases classified as 'not TB' (CRS-negative). Of the 219 cases defined based on the CRS, 69 (31.5%) were confirmed TB and 150 (68.5%) were probable TB cases. The Xpert detected 42/69 (60.9%) patients with confirmed TB and negative culture and 52/69 (75.4%) patients with negative AFB smear results. Among 33 smear and culture positive cases one was not detected by Xpert, and among 23 smear negative and culture positive samples two were not detected by Xpert. The flow chart depicting the inclusion of patients, CRS classification, and laboratory tests are shown in Fig 1.

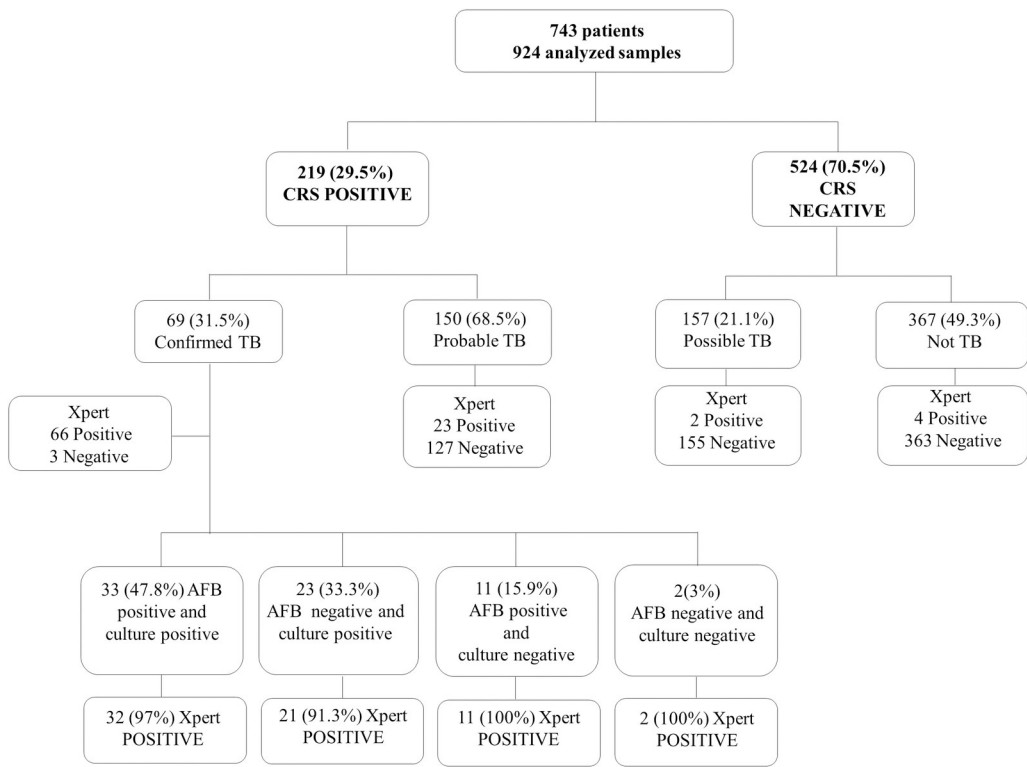

**Fig 1. Flowchart of the patients included and diagnostic classifications.** Indeterminate cases were excluded from the CRS reference standard. All percentages report the proportions of the respective patients, relative to the 743 patients studied.

## Demographic and clinical data

Of 743 patients, 448 (60%) were males and displayed a mean age of 46.7 years (34.4–58.1). Among 364/743 patients who reported contact with TB cases, 59 (16%) described close contact; of these, 34 (57%) had contact with TB patients outside the home. Regarding TB treatment, 339/743 were treated, of which 61 (18%) had TB previously, 18 (29.5%) had abandoned treatment, 31 (50.8%) were cured, and 11 (17.9%) did not provide information at the end of the treatment. A total of 266/743 (35.8%) patients tested positive for HIV, of which 107 (48.8%) cases represented co-infection with TB (Table 2). The lethality rate was 22.4% (167/743), and the cause of death was TB and HIV in 16 (9.6%) and 45 (27%) patients, respectively.

## Xpert MTB/RIF performance in pulmonary and extrapulmonary samples

Out of the 924 samples evaluated, 514 (55.6%) were pulmonary (Table 3). The molecular test detected 23 pulmonary samples that were negative in the smear and culture analyses, representing a 25.6% increase in diagnosis (Fig 2A). Performing the same comparative analysis in extrapulmonary samples (Table 3), 18 out of 410 positive samples could be detected only by molecular testing, showing a 54.5% increase in diagnosis (Fig 2B). Furthermore, Xpert results were available within a maximum of 24 hours (between collection and result), whereas the turnaround time of conventional culture is an average of 3 to 8 weeks.

The performance of the three methods is shown in Fig 3. Considering culture as the gold standard, Xpert showed excellent specificity and sensitivity. It performance surpassed that of the AFB smear in pulmonary and extrapulmonary samples, although with a low positive predictive value (PPV).

**Table 2. Demographic and clinical characteristics of the studied patients according to composite reference standard classification.**

| Characteristics | All study patients (n = 743) n (%) | Confirmed/Probable TB (n = 219) n (%) | Possible TB (n = 157) n (%) | Not TB (n = 367) n (%) |
|---|---|---|---|---|
| **Gender** | | | | |
| Male | 449 (60.4) | 140 (63.9) | 103 (65.6) | 103 (28) |
| Female | 294 (39.6) | 79 (37.6) | 54 (34.4) | 54 (34.4) |
| **Age** | | | | |
| Age <30 years | 105 (14.1) | 17 (7.8) | 24 (15.2) | 64 (17.4) |
| 30–60 years | 426 (57.3) | 161 (73.5) | 96 (61.1) | 169 (46) |
| >60 years | 212 (28.5) | 41 (18.7) | 37 (23.6) | 134 (36.5) |
| **HIV status** | | | | |
| Positive | 266 (35.8) | 107 (48.8) | 70 (44.6) | 89 (24.2) |
| Negative | 477 (64.2) | 112 (51.1) | 87 (55.4) | 278 (75.7) |
| **Other comorbidities** | | | | |
| Yes | 160 (21.5) | 74 (33.8) | 53 (33.7) | 33 (9.0) |
| No | 198 (26.6) | 108 (49.3) | 70 (44.5) | 20 (5.4) |
| Not informed/knew[a] | 385 (51.8) | 37 (16.9) | 34 (21.6) | 314 (85.5) |
| **Specimen type**[b] | | | | |
| Pulmonary | 514 (55.6) | 160 (73) | 109 (69.4) | 157 (42.7) |
| Extrapulmonary | 410 (44.4) | 59 (26.9) | 48 (30.6) | 210 (57.2) |

Other comorbidities include adenocarcinomas, diabetes, hypertension, and autoimmune diseases.

[a]For 385 patients, the data was not available.

[b]All percentages report the proportions of the respective patients relative to the 924 specimens analyzed.

Based on the CRS classification, Xpert sensitivity was very low, but significantly better than that of AFB smear and culture (P < 0.0001), while its specificity was 99.6%. There was no significant difference in the values of PPVs and NPVs between the different methods. The sensitivity of the Xpert test in pulmonary samples was 44.6%, higher than that found in

**Table 3. Pulmonary and extrapulmonary specimens: Comparison AFB smear, culture (LJ), and Xpert assay.**

| Specimen type (n) | AFB | | Culture (LJ) | | Xpert | |
|---|---|---|---|---|---|---|
| **Pulmonary** | + | - | + | - | + | - |
| Sputum (349) | 38 | 311 | 44 | 305 | 70 | 279 |
| Bronchoalveolar lavage (141) | 6 | 135 | 7 | 134 | 12 | 129 |
| Tracheal aspirates (15) | 1 | 14 | 2 | 13 | 3 | 12 |
| Gastric aspirate (9) | 2 | 7 | 2 | 7 | 2 | 7 |
| **Total samples (514)** | 47 | 467 | 55 | 459 | 87 | 427 |
| Specimen type (n) | AFB | | Culture (LJ) | | Xpert | |
| **Extrapulmonary** | + | - | + | - | + | - |
| Cerebrospinal fluid (194) | 0 | 194 | 2 | 192 | 7 | 187 |
| Other body fluids (78)* | 1 | 77 | 2 | 76 | 4 | 74 |
| Biopsies (77) | 2 | 75 | 2 | 75 | 7 | 70 |
| Urine (49) | 3 | 46 | 4 | 45 | 10 | 39 |
| Abscess (9) | 1 | 8 | 2 | 7 | 2 | 7 |
| Pus (2) | 1 | 1 | 1 | 1 | 1 | 1 |
| Lymph node tissue (1) | 0 | 1 | 1 | 0 | 1 | 0 |
| **Total samples (410)** | 8 | 402 | 14 | 396 | 32 | 378 |

*Other body fluids include pericardial, pleural, and synovial fluids.

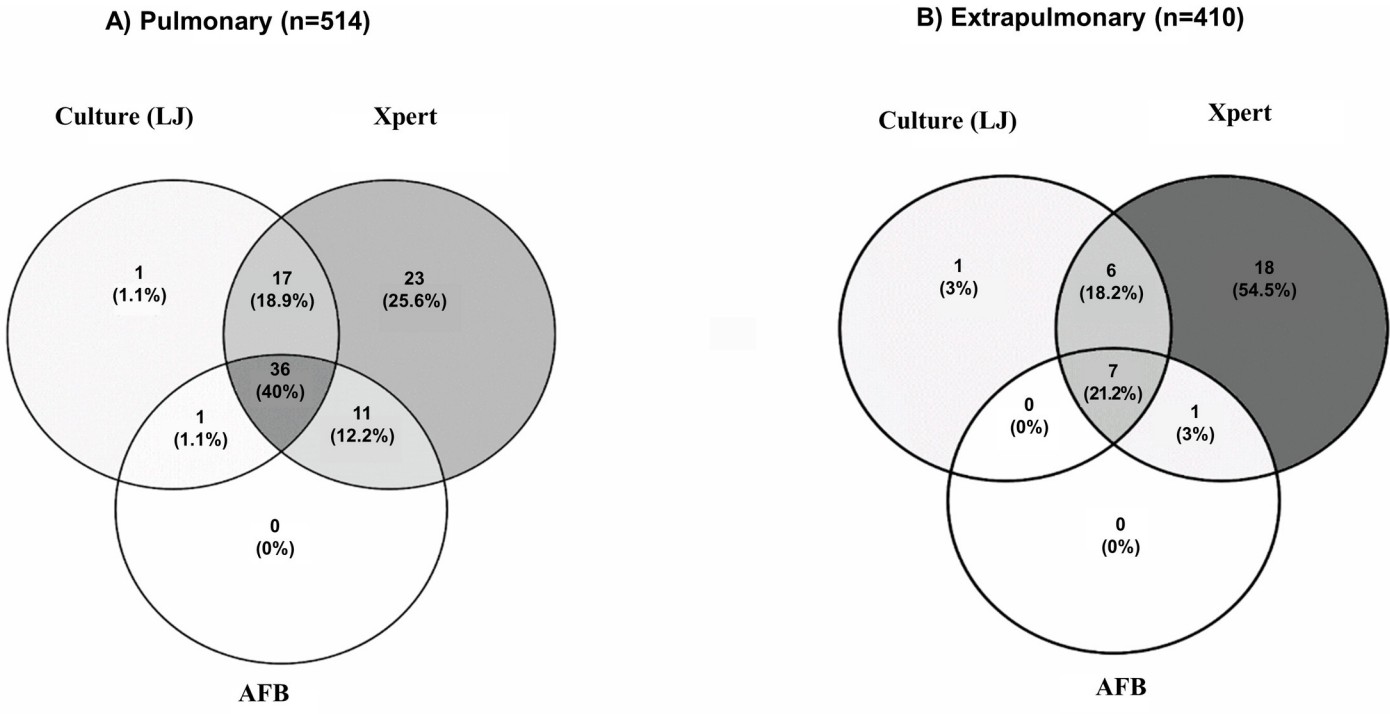

**Fig 2.** Venn diagram showing the relationship between test positivity for Xpert, culture and AFB smears of pulmonary (A) and extrapulmonary (B) samples.

extrapulmonary samples 29.3%. However, specificity and predictive values were similar between these samples sites.

The Xpert automated results can estimate mycobacterial load by measuring the threshold-cycle (Ct), which is a semi-quantitative analysis. Comparison of Xpert Ct values and AFB smear positivity revealed an inverse correlation between them. All Xpert "high load" (Ct < 16) values had a positive smear result. Among the "medium load" (Ct 16–22) values, 21/31 (68%) were smear positive, and those results with "low load" (Ct 22–28) and "very low load" (Ct > 28) only 7/40 (17.5%) and 3/23 (13%) were AFB smear positive, respectively.

The Kappa coefficients obtained in the comparison of Xpert with other laboratory methods using pulmonary and extrapulmonary samples are presented in Table 4. In general, there was a moderate to good agreement, but when analyzing only the results of extrapulmonary samples, a poor agreement was observed between Xpert and AFB results.

Resistance to rifampicin was detected by molecular testing in 7 of the 924 samples. Of those, three were from an abdominal abscess of the same patient with confirmed extrapulmonary TB. In the same samples, rifampicin- and isoniazid-resistant *M. tuberculosis* strains were identified using DST. The remaining samples (four) had culture-negative results, and one of these patients died due to MDR-TB.

## Xpert MTB/RIF performance in HIV patients

Regarding the clinical condition of the 266 HIV patients in this study, 150 (56.6%) presented with current T-CD4+ lymphocyte counts below 200 cells/mm³, and 162 are undergoing antiretroviral therapy (ART). Pulmonary TB was more frequent than extrapulmonary TB in these patients. The Xpert test showed positive results in 42/219 (19.2%) CRS-positive, of which most were those with advanced immunosuppression showing negative culture results.

**Test Performance**

| Gold Standard | Screening Method and sample | PPV (%) | PVN (%) | Sensitivity (CI 95%) | Specificity (CI 95%) | Sensitivity (CI 95%) | Specificity (CI 95%) |
|---|---|---|---|---|---|---|---|
| Culture | Xpert | 55 | 100 | 96 (87.8-100) | 94 (91.8-95.3) | | |
| | AFB | 78 | 97 | 62 (49.8-73.7) | 99 (97.6-99.3) | | |
| | Pulmonary sample *vs* Xpert | 62 | 99 | 91 (87/514) (80-97) | 93 (90.5-95.4) | | |
| | Extrapulmonary sample *vs* Xpert | 41 | 99 | 87 (32/410) (59.5-98.3) | 95 (92.6-97.1) | | |
| CRS | Xpert | 97 | 80.3 | 40.7 (89/219) (34.1-47.6) | 99.6 (98.6-99.5) | | |
| | AFB | 100 | 75 | 20 (44/219) (15-26) | 100 (99.3-100) | | |
| | Culture | 100 | 76.6 | 25 (57/219) (20.2-32.3) | 100 (99.3-100) | | |
| | Pulmonary sample vs Xpert | 97.3 | 54.9 | 44.6 (71/219) (36.9-52.7) | 98 (93.5-99.8) | | |
| | Extrapulmonary vs Xpert | 100 | 52.9 | 29.3 (17/219) (18-42.7) | 100 (92.3-100) | | |

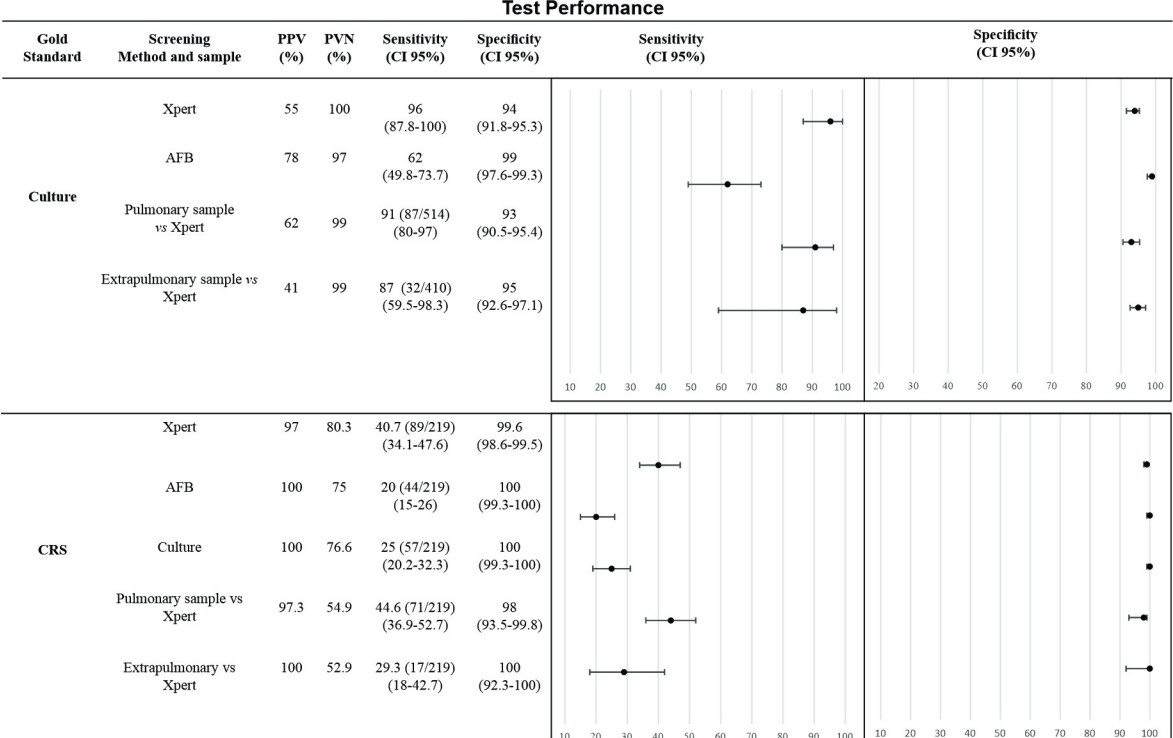

**Fig 3. Gold Standard: Culture results and CRS classification.** The circle in the plot represents the sensitivity and specificity of each diagnostic method, the black line indicates the confidence interval (95% confidence interval). CI, confidence interval; PPV, positive predictive value; NPV, negative predictive value.

A comparison of laboratory methods in different samples (pulmonary and extrapulmonary), derived from HIV-positive and -negative patients, is shown in Fig 4. There was no statistical difference between the populations.

Using CRS as the reference, the sensitivity and specificity of Xpert were the same in both populations (HIV-positive and -negative). When compared to the AFB smear and culture, the molecular test showed higher sensitivity values in both populations, as shown in Fig 5.

## Discussion

In this study, the Xpert assay was found to be a sensitive, specific, and rapid tool for the diagnosis of pulmonary and extrapulmonary TB, including patients with severe

**Table 4. Kappa coefficient of comparison of conventional microbiological methods with Xpert.**

| Method | Samples | Kappa (CI95%) | Criteria |
|---|---|---|---|
| Xpert and AFB | Pulmonary and Extrapulmonary | 0.60 (0.50–0.69) | Moderate |
| Xpert and Culture | Pulmonary and Extrapulmonary | 0.67 (0.55–0.75) | Good |
| Xpert and AFB | Pulmonary | 0.66 (0.56–0.76) | Good |
| Xpert and Culture | Pulmonary | 0.70 (0.61–0.80) | Good |
| Xpert and AFB | Extrapulmonary | 0.38 (0.14–0.62) | Poor |
| Xpert and Culture | Extrapulmonary | 0.54 (0.34–0.73) | Moderate |

Values are expressed in n (CI95%). The criteria applied for the kappa coefficient were < 0.20 = poor; 0.21–0.40 = weak; 0.41–0.60 = moderate; 0.61–0.80 = good; and > 0.80–1.00 = very good.

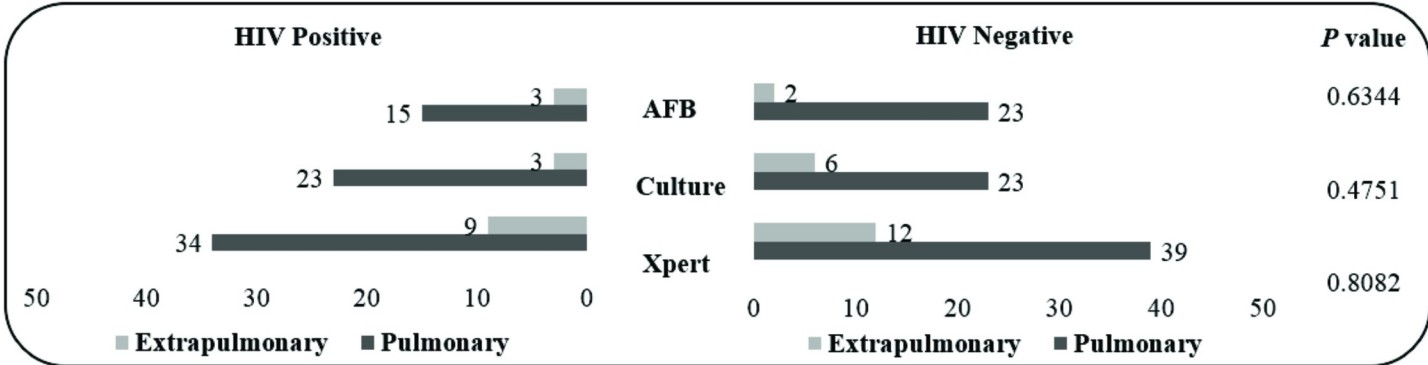

**Fig 4. Number of positive pulmonary and extrapulmonary samples based on the method used in HIV-positive and -negative patients.**

immunodeficiencies who are usually paucibacillary, emphasizing the importance of including molecular testing as an additional tool for the diagnosis of TB.

From the sociodemographic aspects, several studies in Brazil have reported similar findings [23–25], with the pulmonary form being the most prevalent [24]. A favorable outcome of treatment was observed in most new TB cases, which is in line with previous report [25]. Early diagnosis, followed by immediate and appropriate treatment, significantly contributes to TB infection control [26]. Phenotypic tests are limited during the pre-analytical and analytical

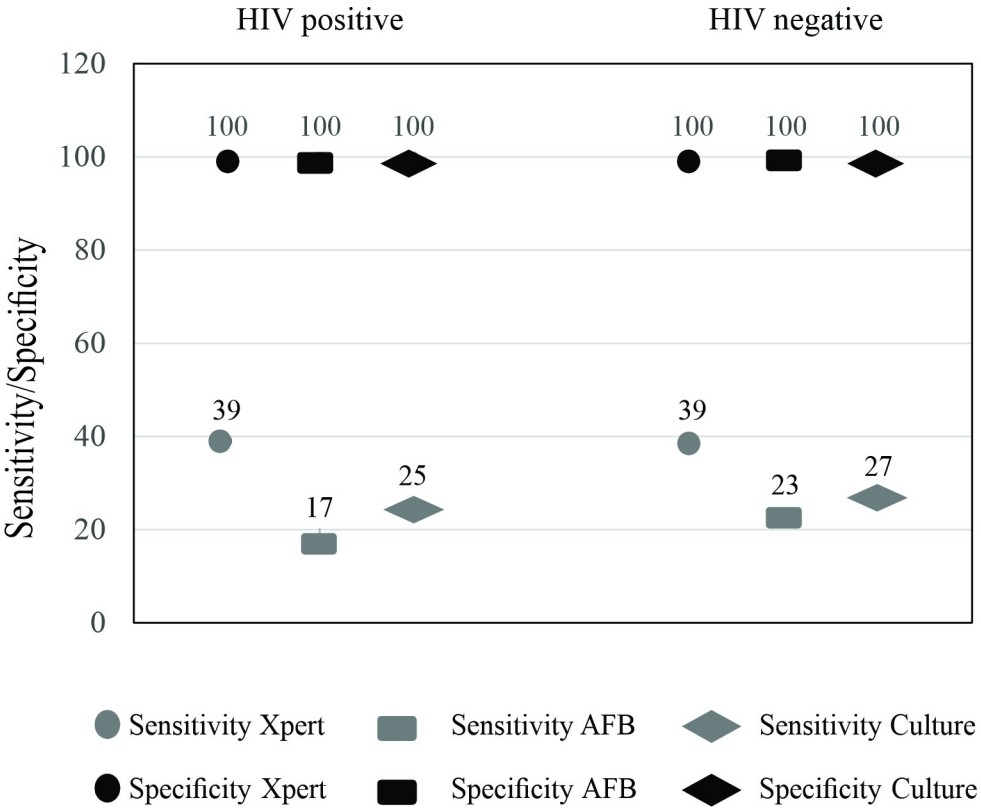

**Fig 5. Sensitivity and specificity of the thee tests Xpert, AFB smear and culture in population HIV positive and not HIV.**

phases [27,28]; therefore, the use of phenotypic tests alone as a reference to determine the accuracy of a molecular test may underestimate the specificity of the method, including the evaluation of resistance to rifampicin [29,30]. CRS offers information that influences treatment decisions; nonetheless, it may produce false-positive results. Therefore, combining two reference standards (culture and CRS) can improve treatment decisions, and ultimately, efficacy [31].

As demonstrated in the present study, the differences of sensitivity may be due to the criteria used as gold standard. High sensitivities are obtained when comparing different laboratory methods; however, when diagnostics were based only on clinical-epidemiological classification, a reduction in the sensitivity was observed. In such cases, the final diagnosis of TB is frequently based on clinical and radiographic findings, which can be the deciding factor influencing the choice of treatment, especially in regions where there is high prevalence of the disease [32]. In addition, such situations occur with paucibacillary patients, and the final confirmation of TB is made via observation of therapy response [21].

Herein, we observed a 25.8% (23/514) increase in microbiological TB diagnosis for pulmonary samples, which were detected only in the molecular test (smear and culture-negative) and a 54.5% (18/410) increase in the extrapulmonary samples. Casela et al. described a diagnostic gain of 59.9% upon comparing the Xpert test with smear results [28] under routine conditions. Other studies showed variable results regarding the diagnostic gain in pulmonary and extrapulmonary samples [18,33–35]. Similar to the findings of Afsar et al., the sensitivity of Xpert in this study varied based on the sample type and the parameters used during analysis. The sensitivity of Xpert compared to culture was slightly higher in the pulmonary than in the extrapulmonary samples [36].

Considering CRS classification, a substantial decrease in sensitivity was observed in Xpert, with it being slightly higher in pulmonary than extrapulmonary samples. The sensitivity reported in this study was lower than that reported in the study by Zeka et al., who reported an overall sensitivity of 70%, 82.3% in pulmonary samples and 52.1% in extrapulmonary samples, but it was evaluated a much smaller number of patients than was evaluated in this study (110 patients) [37]. Previously, a meta-analysis reported a sensitivity and specificity of 59% and 99%, respectively, for Xpert in different regions with an endemic burden in the pulmonary samples [31]. Vadwai et al. reported a sensitivity of 81% in combination with CRS in 283 extrapulmonary samples and observed a good sensitivity for fluid samples, moderate sensitivity for biopsy samples, and low sensitivity for cerebrospinal fluid (CRS) samples [18]. Meanwhile, other studies reported that Xpert displayed a good performance in detecting TB in urine samples, despite a lack of recommendations for these specimens [16,38]. In our study, the Xpert detected 6/13 cases confirmed with genitourinary tuberculosis.

Based on this observation, it has been concluded that the Xpert has good specificity, but limited sensitivity, mainly in extrapulmonary samples. It means, a negative result does not exclude the disease [39]. Moreover, investigation of extrapulmonary TB is complex and the sensitivity of the technique used varies according to the population studied, the quality and quantity of bacillary load in clinical samples, and the laboratory method used [40].

In this study, the PPV and NPV were higher when the results of the molecular tests were compared with the CRS-positive results, as clinical, radiographic, and microbiological parameters were considered for the diagnosis of TB. The PPV depends on the clinical probability of the disease; however, the studied population did not have a high probability of TB, given that it represented a broad population. In contrast, Marouane et al. obtained a sensitivity of 84.7% and PPV of 94.3% in extrapulmonary samples from patients with a high probability of TB when comparing the Xpert MTB/RIF test to the Ziehl–Neelsen fluorescence and liquid culture tests [11].

The Xpert test was positive in all patients with positive smears, indicating that results of both methods correlate well. Similar results were previously reported [15,41], wherein comparison of the GeneXpert cycle threshold with the smear results showed that most of the "low" and "very low" Ct values in the molecular test were negative in the smear, thus proving its value in the early identification of bacilli before the culture result [28]. When compared with CRS-positive results, both AFB and culture had a similar sensitivity of 20% and 25% respectively.

Discordant results between positive Xpert and negative culture in patients with confirmed TB was observed in 13 of the 69 patients. These discordant results were observed in paucibacillary samples, where cycle threshold values were medium and very low. It is probable that the bacterial count was below the limit of detection in the culture, the decontamination process was inefficient or due to nonviable mycobacterium [42].

Studies have indicated that good sensitivity and specificity may be obtained in the detection of the rifampicin resistance mutation among positive and negative [43,44] smear samples; however, the sensitivity of the molecular test may be limited in paucibacillary samples. In sites with low incidence of TB resistance, silent mutations in the *rpoB* gene are common; although they do not alter the properties of the encoded proteins, they may have false positive rifampicin resistance in the Xpert test [29,44].

In this study, all three samples identified as resistant using conventional methods were also found to be resistant using Xpert, similar to findings previously reported by Afsar et al. [36]. A recent study by Huo and colleagues found that cases with very low bacterial load were more likely to be misdiagnosed with Xpert resistance to RIF. Notably, in the present study, four quantifications of resistant samples that were "very low" or "low" were positive for Xpert and negative for culture, which justifies further investigations [45].

The WHO recommends confirming cases of resistance detected in the Xpert test by phenotypic or other genotypic methods (Line-Probe Assay or nucleotide sequencing) and to solve any discordant rifampicin susceptibility results [29] using a new sample to increase the sensitivity of rifampicin resistance diagnosis, especially in Brazil, which has a low prevalence of resistant TB [46,47].

The Xpert test can present false-positive results in patients who have had previously active TB, as genetic material from dead bacilli remains detectable in the sputum of patients after treatment [28]. Some studies have concluded that Xpert detects MTB for up to five years after treatment [48,49]. Therefore, in these cases, the diagnosis of active TB should be performed by smear and culture tests, whereas the molecular test is recommended for early diagnosis and is not indicated for monitoring therapeutic response [50]. In our study, there were five patients who did not have active TB but had previously undergone treatment, and one patient was under treatment at the time of the molecular test and were considered "not TB". We found that two hemorrhagic samples of alveolar bronchus lavage showed negative results in Xpert. According to the manufacturer Cepheid®, endogenous substances can interfere, such as blood, leukocytes, respiratory tract cells, mucin, human DNA, gastric acid and some medications [51].

Of the patients included in this study, 35.8% (266/743) were HIV-positive and most of them already met the criteria for AIDS diagnosis. Low T-CD4 lymphocyte count is associated with advanced stage of TB/HIV coinfection along with immunological impairment and critical clinical conditions [42–45]. Comparing the results of the Xpert in HIV-positive and -negative populations, no significant differences were observed in the test performance. In contrast, Brum et al. reported that the Xpert test is sensitive and specific in cases of pulmonary TB among populations living with HIV, thereby showing diagnostic and treatment benefits, especially in cases with negative smear results [52,53]. Auld et al. concluded that the molecular test

contributed to an increase in TB-MDR detection and observed a decline in the empirical treatment of this population, contributing to timely diagnosis and treatment [7].

The HIV population in this study had higher chances of morbidity and mortality based on the low T-CD4 cell count and detectable viral load; the Xpert test could detect TB infection in cases presenting negative cultures predominantly in this population, as clinical and laboratory characteristics of these infections are usually atypical and difficult to diagnose [52].

The Xpert test is valuable for the initial diagnosis of tuberculosis, especially in its extrapulmonary form, which is more frequent in HIV clinics. Furthermore, rapid diagnosis and immediate treatments that overcome the limitations of conventional tests are essential [53], especially in patients with low T-CD4+ lymphocyte counts, undetectable viral load, irregular use of ART, advanced age and male gender, in whom the chances of death from TB are high [54,55].

Introduction of the Xpert test in the laboratory routine at a low incidence site showed an increase in the diagnosis of active TB compared to phenotypic tests, in addition to being an easy and useful tool to obtain rapid and reliable results with high sensitivity and specificity in variable populations and specimens. Early diagnosis allows immediate initiation of timely treatment and thereby contributes substantially to the control of morbidity and mortality and the risk of TB transmission, especially in populations living with HIV having severe immunological impairment. For proper management of TB/HIV co-infection, Xpert is considered a useful tool for diagnosing extrapulmonary and paucibacillary TB infections, as the symptoms in these patients are often atypical. In the present study, positivity in extrapulmonary samples was 54.5%, showing a diagnostic gain in cases of confirmed extrapulmonary TB.

The main limitations of the current study were difficulties in obtaining adequate sample volumes, especially for extrapulmonary samples, preventing centrifugation for determining the bacillary load and requiring new collection for confirmation of resistance in some cases. Additionally, there were limited clinical secondary data.

In summary, our study highlights the benefits of Xpert MTB/RIF in different clinical settings combined with the clinical history and results of other diagnostic tests, especially in retreatment tuberculosis cases. Further, clinical and epidemiological studies evaluating the performance of the new version of Xpert® MTB/RIF Ultra in extrapulmonary samples, HIV population, and children are warranted to define the contribution of the new version in this critical sample and population.

## Acknowledgments

We thank the bacteriology laboratory of the Diagnostic Support Unit (ULAC), the epidemiology and infectology sector of the Hospital de Clínicas of the Federal University of Paraná (CHC/UFPR). We also thank the Municipal Health Secretariat (SMS) Curitiba/Paraná for the access to databases for clinical and epidemiological data collection. We thank the Central Laboratory of Paraná (LACEN), Paraná, Brazil, for conducting the antimicrobial susceptibility test.

## Author Contributions

**Conceptualization:** Sonia Mara Raboni, Gislene Maria Botão Kussen, Clea Elisa Lopes Ribeiro, Libera Maria Dalla Costa.

**Data curation:** Ana Paula de Oliveira Tomaz, Sonia Mara Raboni, Gislene Maria Botão Kussen, Keite da Silva Nogueira, Clea Elisa Lopes Ribeiro.

**Formal analysis:** Ana Paula de Oliveira Tomaz, Keite da Silva Nogueira.

**Investigation:** Ana Paula de Oliveira Tomaz.

**Methodology:** Ana Paula de Oliveira Tomaz, Sonia Mara Raboni, Gislene Maria Botão Kussen, Libera Maria Dalla Costa.

**Supervision:** Libera Maria Dalla Costa.

**Validation:** Ana Paula de Oliveira Tomaz.

**Writing – original draft:** Ana Paula de Oliveira Tomaz.

**Writing – review & editing:** Ana Paula de Oliveira Tomaz, Sonia Mara Raboni, Gislene Maria Botão Kussen, Keite da Silva Nogueira, Libera Maria Dalla Costa.

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
