## [Decision Letter · Decision Letter 0]

26 Oct 2020

PONE-D-20-29217

Xpert® MTB/RIF diagnostic test for pulmonary and extrapulmonary tuberculosis in immunocompetent and immunocompromised patients: benefits and experiences over 2 years in different contexts

PLOS ONE

Dear Dr. TOMAZ,

Thank you for submitting your manuscript to PLOS ONE. After careful consideration, we feel that it has merit but does not fully meet PLOS ONE’s publication criteria as it currently stands. Therefore, we invite you to submit a revised version of the manuscript that addresses the points raised during the review process.

The findings of the study has important implications in TB diagnostics and control. But the manuscript needs work and improvements in the clarity, especially distinction in the results between Xpert and and Xpert ultra. I hope the authors consider the reviewers comments useful in improving the manuscript and I look forward for your revised version.

We look forward to receiving your revised manuscript.

Kind regards,

Padmapriya P Banada, PhD

Academic Editor

PLOS ONE

Journal Requirements:

Reviewers' comments:

Reviewer's Responses to Questions

**Comments to the Author**

1. Is the manuscript technically sound, and do the data support the conclusions?

Reviewer #1: Yes

Reviewer #2: Partly

2. Has the statistical analysis been performed appropriately and rigorously? 

Reviewer #1: Yes

Reviewer #2: I Don't Know

3. Have the authors made all data underlying the findings in their manuscript fully available?

Reviewer #1: Yes

Reviewer #2: Yes

4. Is the manuscript presented in an intelligible fashion and written in standard English?

Reviewer #1: Yes

Reviewer #2: Yes

5. Review Comments to the Author

Reviewer #1: Summary: This prospective observational study aimed to evaluate the performance of Xpert MTB/RIF in different contexts as listed below

(i) laboratory and clinical/epidemiological diagnosis;

(ii) HIV-positive and -negative populations; and

(iii) type of specimens: pulmonary and extrapulmonary.

The Xpert MTB/RIF results were compared with those of Ziehl–Neelsen stained smears and BAAR cultures on solid L.J medium for specimens belonging to clinically suspected TB cases during a two-year period. Total, 924 specimens (pulmonary 514/55.7% and extrapulmonary 410/44.3%) from 743 patients were evaluated. Among these, according to clinical criteria, 101 (13.6%) were classified as confirmed TB cases, 36 (4.8%) probable TB and 606 (81.6%) improbable TB cases. Overall sensitivity, specificity, positive predictive value, and negative predictive value results of Xpert MTB/RIF compared to those of the L.J cultures were 96%, 94%, 57%, and 99%, respectively. The Xpert MTB/RIF test showed an increase in the detection of 13

(35.1%) of 37 confirmed cases of TB in patients with advanced immunosuppression showing negative culture.

Recommendation: Accept with major revisions as described below

Comments:

1. Phenotypic culture confirmatory method used in this study (L.J culture) is conventional and very suboptimal reference standard for detection of extra-pulmonary and paucibacillary TB. In HIV-positive patients, the diagnosis of pulmonary tuberculosis (PTB) is a challenging task due to the paucibacillary nature of the infection, which greatly reduces the effectiveness of smear microscopy and conventional L.J. culture techniques that are used as a phenotypic reference standard to confirm TB in this study. So, I would recommend authors to use composite reference standard (CRS) as a gold standard to evaluate the true diagnostic potential of Xpert MTB/RIF assay and other diagnostic tests for detection extra-pulmonary and paucibacillary TB. Calculation of sensitivity, specificity, PPV and NPV of all laboratory tests against CRS+ and CRS - patients would give more realistic picture.

2. Clarity in writing and flow need to be improved especially in abstract, methods, results and discussion sections.

Reviewer #2: This study by Tomaz A and colleagues compares Xpert MTB/RIF with smear and culture, and against a clinical/laboratory classification for pulmonary TB, extrapulmonary TB, and TB in persons living with HIV over a 2 year period. The study adds to a large body of literature describing experience with Xpert in these populations. The authors conclude that Xpert leads to increased diagnosis of active TB cases compared to phenotypic tests.

Suggest that the comparators and reference standard for each cohort are explicitly outlined in the methods section.

Lines 142, 149-150 mentions outcomes –how long were the participants followed?

Differences in time to diagnosis may be an opportunity to look at operational advantages of Xpert versus culture – was this captured?

Did the local providers or study investigators make the determination of confirmed, probable and improbable TB for each patient? Were these reference determinations made blinded to the study Xpert, culture, and smear results?

Along these lines, were Xpert MTB/RIF results shared with providers? If so, would Xpert have played a role in provider classification of improbable, probable, and confirmed TB?

Consider including a baseline characteristics table to elucidate proportions and overlap of characteristics such as pediatric, extrapulmonary, HIV, demographic, recent TB treatment, smear positivity, confirmed/probable/improbable

The study procedures are not entirely clear. How did obtaining TB case records (line 84) fit into the prospective enrollment? Were these case records used for recruitment or follow-up? Please describe what procedures took place during the study encounter (how many samples were collected, etc).

Samples can be heterogenous and majority of patients had smear, culture, and Xpert performed on separate samples which can lead to increased discordancy between these tests. Please comment how far apart the sample collections took place particularly in cases where Xpert was compared with culture on separate samples. Please confirm no interval treatment took place between those samples.

Line 164-65: Please clarify how Xpert MTB/RIF showed the best accuracy for confirmed TB when final classification of confirmed and improbable TB were based on phenotypic tests culture and/or sputum smear (line 96-97)?

Lines 139-142: Along these same lines, please clarify how could 36 of 42 culture-negative patients be classified as confirmed TB if the SINAN definition was used?

Fig 1. Consider providing the Venn diagram in context of confirmed, probable, and improbable TB.

Fig 2 and Fig 3. Please define which is element in each comparison is the reference standard.

Fig. 2 legend: "TB confirmed (sick) as probable and improbable (not sick)." Unclear what this means.

Almost half of the patients evaluated were treated for TB (line 134) – how recent were these treatment courses? Even treatment over 6 months prior [Dorman et al] (and even years ago along some lines of evidence) are implicated to result in remnant nucleic acids which are picked up by Xpert but not culture leading to discordancy in favor of Xpert (Xpert-positive, culture-negative).

Lines 297-300: Both authors have pointed out…its [Ultra’s] performance is comparable with that of the Xpert MTB/RIF…

On the contrary - Ultra’s sensitivity was found to be superior to that of Xpert in patients with smear-negative TB and HIV, which was the target population of Ultra that Xpert MTB/RIF too frequently missed [Dorman et al, Lancet 2017]. There is generally little difficulty for nucleic acid amplification tests to pick up smear-positive TB.

6. PLOS authors have the option to publish the peer review history of their article (what does this mean?). If published, this will include your full peer review and any attached files.

Reviewer #1: **Yes: **Dr Shubhada Vasudeo Shenai

Reviewer #2: No

---

## [Author Response · Author response to Decision Letter 0]

15 Dec 2020

December 09th, 2020

Dr. Joerg Heber

Editor-in-Chief 

PLOS ONE 

Dear Dr. Heber:

I, along with my coauthors, wish to submit the revised manuscript entitled "The Xpert® MTB/RIF diagnostic test for pulmonary and extrapulmonary tuberculosis in immunocompetent and immunocompromised patients: benefits and experiences over 2 years in different contexts" (PONE-D-20-29217) by Tomaz et al. for publication in PLOS ONE. 

We would like to thank you and the reviewers for your thoughtful suggestions and insights which have significantly helped us to improve our manuscript. We have carefully revised the manuscript (the revisions are highlighted in yellow for your convenience) to address the major concerns raised and hope that our revised manuscript meets your standards and will be reconsidered for publication in PLOS ONE. 

We have also provided our point-by-point response (in red color) to all comments raised and hope that our replay addressed all your concerns.

Should you have any further questions, please contact us.

Thank you for your consideration. I look forward to hearing from you. 

Sincerely,

Libera Maria Dalla Costa

Faculdades Pequeno Príncipe (FPP), Curitiba, Paraná, Brazil

Instituto de Pesquisa Pelé Pequeno Príncipe (IPPPP), Curitiba, Paraná, Brazil

lmdallacosta@gmail.com

libera.costa@professor.edu.br

 

Title: The Xpert® MTB/RIF diagnostic test for pulmonary and extrapulmonary tuberculosis in immunocompetent and immunocompromised patients: benefits and experiences over 2 years in different contexts 

Reviewer #1 

Comments:

1. Phenotypic culture confirmatory method used in this study (L.J culture) is conventional and very suboptimal reference standard for detection of extra-pulmonary and paucibacillary TB. In HIV-positive patients, the diagnosis of pulmonary tuberculosis (PTB) is a challenging task due to the paucibacillary nature of the infection, which greatly reduces the effectiveness of smear microscopy and conventional L.J. culture techniques that are used as a phenotypic reference standard to confirm TB in this study. So, I would recommend authors to use composite reference standard (CRS) as a gold standard to evaluate the true diagnostic potential of Xpert MTB/RIF assay and other diagnostic tests for detection extra-pulmonary and paucibacillary TB. Calculation of sensitivity, specificity, PPV and NPV of all laboratory tests against CRS+ and CRS - patients would give more realistic picture.

Thank you for your comment and evaluation. Methods including microscopic analysis of smears and conventional culture techniques are not ideal reference standards to confirm TB. We used an algorithm to categorize the patients. First, we identified whether patients were notified at SINAN – the Notifiable Diseases Information System; then, we evaluated the laboratory results, clinical and radiologic findings, and treatment response to categorize the patients into 4 groups: (i) “Confirmed TB”, AFB-positive/culture-positive and AFB-negative/culture-positive patients; (ii) “Probable TB”, (AFB/culture-negative patients showing clinical signs/symptoms of TB, radiological findings and/or histology suggestive of TB); (iii) “Possible TB”, patients not included in the previous groups and patients showing only clinical signs/symptoms of TB but without any records for the treatment of TB; (iv) “Not TB”, negative culture and negative results for all other tests and patients not registered in the TB database (non-TB treatment) database, as described by Vadwai et al [1]. 

Following the analysis of operational characteristics, tests were performed again using the composite reference standard (CRS), as a gold standard, where “confirmed TB” and “probable TB” were classified as CRS+ cases and “Possible TB” and “not Tb” as CRS-.

1. V. Vadwai, C. Boehme, P. Nabeta, A. Shetty, D. Alland, C. Rodrigues. Xpert MTB/RIF: a new pillar in diagnosis of extrapulmonary tuberculosis? J Clin Microbiol. 2011; 49: 2540-2545.

1. Clarity in writing and flow need to be improved especially in abstract, methods, results and discussion sections. 

The manuscript was reviewed and re-written for clarity.

2. Abstract: Xpert® MTB/RIF assay was first approved for diagnosis of pulmonary tuberculosis and RIF resistance by WHO in 2010. Revised guidelines were published by WHO in 2013 with recommendations for diagnosis of pediatric and extra-pulmonary TB especially CSF, tissue and lymph-node samples. Authors should make these corrections.

The Abstract was revised accordingly, in 2014, the GeneXpert MTB/RIF assay, a rapid tool for diagnosis of tuberculosis (TB), was implanted in Brazilian Unified Health System. The explanation of the guidelines in the introduction was revised for better clarity. (Line 61-66). 

3. Material & Methods: 

a) Study design, patient population, and samples, Lines 77-79

One of the exclusion criteria was patient’s samples that did not yield AFB culture and/or smear results. Authors should elaborate on this with reasons as the traditional phenotypic methods (AFB smear and L.J culture) used in this study itself have many limitations especially while detecting paucibacillary TB from extrapulmonary samples &/or samples collected from HIV patients etc. Further, it would be interesting to know issues related to unavailability of AFB smears results.

Thank you for your comment and suggestion. The reasons for the unavailability of the results are explained in the revised manuscript. (Lines 82-84)

The unavailability of the culture results was due to the inefficient decontamination process, which interrupted the culture due to proteolysis of the medium. Identified mycobacteria from the non-TB group (MNT). We also received an insufficient volume of samples for all laboratory tests.

b) Study design, patient population, and samples, Line 100

Symptomatic patients without any established diagnosis are classified as improbable TB group. Means all the patients included in this group have clinical symptoms and/or signs suggestive of TB but radiology, ZN smear, AFB culture and other tests results are negative for TB but it. Further, phenotypic & other conventional reference methods (AFB smear and/or L.J culture, radiology, histopathology/cytology, etc.) utilized this study have known limitations in detection of TB especially paucibacillary TB from extrapulmonary specimens. Therefore, considering limitations of technology used as a reference standard, possibility of TB disease cannot be eliminated in this group of symptomatic patients. I would recommend changing the terminology improbable TB group as “possible TB”. 

We have revised the terminology as “possible TB”, as per your suggestion. (Lines 123)

c) Laboratory Methods: Line 109-111

Xpert MTB/RIF assay protocol for sample processing for urine to be described as it is not recommended by manufacturers as well as WHO.

One of our study objectives was to evaluate the performance of the Xpert MTB/RIF assay using non-pulmonary samples. Information on the processing of these samples has been mentioned in the materials and methods section. (Lines 96-98)

2. Results: 

a) Study population: Lines 129 -130

Authors mentioned that, of the total, 743 patients included in the study; 616 patients underwent BAAR (smear), culture, and Xpert MTB/RIF tests with a single sample, whereas 310 patients underwent these tests using multiple samples. If we add these two groups of patients (616 with single sample and 310 with multiple samples) then total number of patients are 926 not 743. Kindly clarify number of patients and total number of samples. 

We have revised and corrected the number of patients accordingly. (Lines 154-156)

b) Study population: Lines 131 to 142 

For better understanding, all the patient’s details can be incorporated in a table format. Flow chart with total number of patients & samples, clear data on number of samples excluded, among remaining samples how many collected from of HIV positive and HIV negative patients, number of pulmonary and extra-pulmonary specimens etc. to be included.

We incorporated the information requested with (Figure 1 p.8); in addition, (table 2 p.9) which lists patient’s details, was also included.

c) Comparison against composite reference standards (CRS): 

In result section, table comparing results of ZN smear, L.J culture, Xpert MTB/RIF assay against composite reference standards (CRS) to be included. Sensitivity, specificity, PPV & NPV of these tests against CRS can be included in the same table. 

The analysis of sensitivity, specificity, PPV, and NPV of these tests was recalculated using the composite reference standard (CRS), as a gold standard, as suggested. (Figure 3 p.11).

3. Discussion: Discussion should be more precise and to the point. Discussion on Xpert ultra (lines 291 -305) is unnecessary. Authors should discuss their results in detailed and compare it with other relevant published data. 

Thank you for your suggestion; accordingly, we have excluded this point from the Discussion.

a) Comparison of Xpert CT values with AFB smear: Line 258-260

Table showing comparison of Xpert CT values with ZN smear gradation among different patient population (pulmonary & extra-pulmonary, HIV+ & HIV- etc.) can be incorporated. Further, Xpert CT values can be comparted against L.J culture turnaround time in days among different patient population. 

Thank you for your suggestion. Unfortunately, we had no access to such data; this was therefore included as a limitation in the Discussion section. (Lines 394-397).

b) In this study, there were only 7 RIF resistant cases. Of these three are form same patients. So only 7 samples collected from 5 patient won’t be enough number of samples to discuss on performance of Xpert MTB/RIF assay for detection of RIF resistance. 

Thank you for your comment; we agree with your assessment. This information was included in the Discussion section (Lines 345-350).

c) False RIF resistance calls (n=4): line 273-274

Few low-level RIF resistant mutants are phenotypically susceptible when DST is performed using liquid culture systems like BACTEC 460 TB and MGIT 960 TB system. So further confirmation of four Xpert resistant, phenotypically susceptible TB strains by other molecular methods like Line Probe Assay or sequencing is recommended for confirmation.

Thank you for your comment. These samples were not sequenced; but this information was included in the Discussion section. (Line 352). However, our study did not aim to compare the resistant strains obtained following in the application of Xpert with those obtained following the use of other methods.

d) References: All references should be written properly as per PLOS One guidelines. Journal name & proper page numbers missing in few references. Please recheck and make necessary corrections. 

All references were revised and corrected accordingly.

 

Reviewer #2: 

1. This study by Tomaz A and colleagues compares Xpert MTB/RIF with smear and culture, and against a clinical/laboratory classification for pulmonary TB, extrapulmonary TB, and TB in persons living with HIV over a 2 year period. The study adds to a large body of literature describing experience with Xpert in these populations. The authors conclude that Xpert leads to increased diagnosis of active TB cases compared to phenotypic tests.

Suggest that the comparators and reference standard for each cohort are explicitly outlined in the methods section.

The manuscript was revised, and cases were reclassified based on the composite reference standards (CRS). This information was included in the Materials and Methods section Clinical data and TB case definitions (Line 116-126).

2. Lines 142, 149-150 mentions outcomes –how long were the participants followed?

The participants were followed-up for at least two years, or until death. We have included this information in the Materials and Methods section. (Lines 105-106).

3. Differences in time to diagnosis may be an opportunity to look at operational advantages of Xpert versus culture – was this captured?

Thank you for your suggestion. Samples were collected from institutions in the city of Curitiba and metropolitan region, which arrived at our laboratory at different times; therefore, this is not the ideal study for this type of analysis.

4. Did the local providers or study investigators make the determination of confirmed, probable and improbable TB for each patient? Were these reference determinations made blinded to the study Xpert, culture, and smear results?

Following data collection, confirmed, probable, possible, and non-TB statuses for each patient was confirmed through lab tests, based on clinical, radiological/histological data provided in the SINAN database.

5. Along these lines, were Xpert MTB/RIF results shared with providers? If so, would Xpert have played a role in provider classification of improbable, probable, and confirmed TB?

All laboratorial results (Xpert MTB/RIF) have been shared with clinicians and used to provide a classification of improbable, probable, and confirmed TB.

6. Consider including a baseline characteristics table to elucidate proportions and overlap of characteristics such as pediatric, extrapulmonary, HIV, demographic, recent TB treatment, smear positivity, confirmed/probable/improbable

Table 2, which lists the patients’ information, was included.

7. The study procedures are not entirely clear. How did obtaining TB case records (line 84) fit into the prospective enrollment? Were these case records used for recruitment or follow-up? Please describe what procedures took place during the study encounter (how many samples were collected, etc).

A detailed description of the recruitment process and data collection was included in the revised manuscript “Study population”. (Line 153). We have also included a Flow chart of the number of patients included in the study, their diagnostic classification, and the number of samples analyzed. (Figure 1 p.8).

8. Samples can be heterogenous and majority of patients had smear, culture, and Xpert performed on separate samples which can lead to increased discordancy between these tests. Please comment how far apart the sample collections took place particularly in cases where Xpert was compared with culture on separate samples. Please confirm no interval treatment took place between those samples.

All laboratory analysis (AFB smears, conventional cultures, and molecular tests) were conducted using the same sample. This information was included in the Materials and Methods section. (Line 99-101).

9. Line 164-65: Please clarify how Xpert MTB/RIF showed the best accuracy for confirmed TB when final classification of confirmed and improbable TB were based on phenotypic tests culture and/or sputum smear (line 96-97)?

Final classification of confirmed and improbable TB was based on all phenotypic tests analyzed altogether. Xpert MTB/RIF conducted individually was better than the culture or smear tests for diagnosing TB. We have included a comment regarding this aspect in the revised manuscript. (Figure 1 p.8)

10. Lines 139-142: Along these same lines, please clarify how could 36 of 42 culture-negative patients be classified as confirmed TB if the SINAN definition was used?

We have used the SINAN final classification, which included other clinical, radiologic, and microbiology tests. In addition, tuberculosis cases are reported only after diagnostic confirmation, suspected cases are not included in Sinan database.

11. Fig 1. Consider providing the Venn diagram in context of confirmed, probable, and improbable TB.

Accordingly, this information was incorporated in (Figure 1 p.8) in the revised manuscript.

12. Fig 2 and Fig 3. Please define which is element in each comparison is the reference standard.

We have changed the case definition using the CRS and included this information in (Figures 3 p.11, 4 p.12, and 5 p.13).

13. Fig. 2 legend: "TB confirmed (sick) as probable and improbable (not sick)." Unclear what this means.

We have revised the legend using the CRS criteria.

14. Almost half of the patients evaluated were treated for TB (line 134) – how recent were these treatment courses? Even treatment over 6 months prior [Dorman et al] (and even years ago along some lines of evidence) are implicated to result in remnant nucleic acids which are picked up by Xpert but not culture leading to discordancy in favor of Xpert (Xpert-positive, culture-negative).

All patients who have undergone a recent TB treatment (2 years) were considered as TB negative. We have included this information while describing the diagnostic criteria in the methodology section. 

15. Lines 297-300: Both authors have pointed out…its [Ultra’s] performance is comparable with that of the Xpert MTB/RIF…

On the contrary - Ultra’s sensitivity was found to be superior to that of Xpert in patients with smear-negative TB and HIV, which was the target population of Ultra that Xpert MTB/RIF too frequently missed [Dorman et al, Lancet 2017]. There is generally little difficulty for nucleic acid amplification tests to pick up smear-positive TB.

Thank you for your comment and suggestion; however, upon accepting the suggestion of reviewer #1, we have excluded this point of discussion. Thank you

---

## [Decision Letter · Decision Letter 1]

3 Feb 2021

The Xpert® MTB/RIF diagnostic test for pulmonary and extrapulmonary tuberculosis in immunocompetent and immunocompromised patients: benefits and experiences over 2 years in different clinical contexts

PONE-D-20-29217R1

Dear Dr. TOMAZ,

We’re pleased to inform you that your manuscript has been judged scientifically suitable for publication and will be formally accepted for publication once it meets all outstanding technical requirements.

Kind regards,

Padmapriya P Banada, PhD

Academic Editor

PLOS ONE

Additional Editor Comments (optional):

Thank you for your resubmission and addressing the concerns raised by the reviewers. I am happy to recommend this for publication.

Reviewers' comments:

Reviewer's Responses to Questions

**Comments to the Author**

1. If the authors have adequately addressed your comments raised in a previous round of review and you feel that this manuscript is now acceptable for publication, you may indicate that here to bypass the “Comments to the Author” section, enter your conflict of interest statement in the “Confidential to Editor” section, and submit your "Accept" recommendation.

Reviewer #1: All comments have been addressed

Reviewer #3: (No Response)

2. Is the manuscript technically sound, and do the data support the conclusions?

Reviewer #1: Yes

Reviewer #3: Yes

3. Has the statistical analysis been performed appropriately and rigorously? 

Reviewer #1: Yes

Reviewer #3: Yes

4. Have the authors made all data underlying the findings in their manuscript fully available?

Reviewer #1: Yes

Reviewer #3: Yes

5. Is the manuscript presented in an intelligible fashion and written in standard English?

Reviewer #1: Yes

Reviewer #3: Yes

6. Review Comments to the Author

Reviewer #1: Many of the reviewer’s comments have been addressed in the revised version, however, few minor corrections are recommended below for further improvement,

1. Table 1: Actual numbers of confirmed, probable, possible, and non-TB patients can be included in brackets (n=…) along with patient’s number for AFB smear, culture, Radiology and other tests data.

2. Line 160 & 162: Among the confirmed TB cases, Xpert detected 32 of the 33 positive samples in culture and AFB smear, 21 of the 23 positive samples in culture, and all positive samples in AFB smear. Kindly reframe this statement to simplify for better understanding.

Recommendation: Among 33 smear and culture positive cases one was not detected by Xpert, and among 23 smear negative culture positive samples 2 were not detected by Xpert. Authors can describe possible causes of these 3-culture positive Xpert negative (false negatives) controversial cases in discussion.

3. Figure 2: Kindly add total no of pulmonary and extra-pulmonary samples in brackets e.g. A) Pulmonary (n= ), and B) Extra Pulmonary (n= ). Further, as described in Fig 2, there were 3 culture positive Xpert negative cases among pulmonary samples and one culture positive Xpert negative case among extra-pulmonary samples. So total 4 culture positive Xpert negative cases. However, in the text (line 160-162) authors have mentioned only three false negative (culture positive Xpert negative) from pulmonary cases instead of total four (pulmonary + extra-pulmonary) cases. Kindly recheck and make corrections wherever required.

4. Confirmed and Probable TB cases data can be separated in table 2, and can be discussed separately in the text.

5. References: All references should be written properly as per PLOSOne guidelines. Please recheck all references and make necessary corrections.

Reviewer #3: This manuscript describes the performance characteristics of Xpert MTB/RIF (Xpert) using a composite reference standard and solid culture (Lowenstein Jensen) in HIV positive and HIV negative individuals. Most of the previous comments have been addressed, or, reasonable justification has been provided by authors to support current presentation of findings. Please find additional comments / recommendations for your consideration.

Specific comments:

Line 65-66: Syntax issue noted, kindly consider rephrasing.

Line 83: Please confirm this is in reference to individuals without a valid (definitive positive or negative) culture result.

Line 94: Reviewer #1 raised questions regarding sample processing for Xpert, however it is still not clear if samples used for Xpert testing were processed sediments or if the samples were split before processing for Xpert and culture (all sample types that were tested by both methods). Please confirm.

Line 234: Typographical error noted, please remove "a"

Line 266: Typographical error noted, please change "tree" to "three"

Line 315: Syntax issue noted, kindly consider rephrasing

Line 334-338: The detection of nonviable mycobacterium could also result in Xpert (+) / culture (-) results, please consider adding.

Line 361-363: Please confirm CRS category for patients described (quoted from manuscript: "five patients who did not have active TB but had previously undergone treatment, and one patient was under treatment at the time of the molecular test")

Line 374-382: The increase in EPTB in HIV negative individuals as compared to HIV positive individuals is an interesting observation, this could be further explored in this section of the discussion.

7. PLOS authors have the option to publish the peer review history of their article (what does this mean?). If published, this will include your full peer review and any attached files.

Reviewer #1: **Yes: **Dr Shubhada Shenai

Reviewer #3: No

---

## [Editor Report · Acceptance letter]

10 Feb 2021

PONE-D-20-29217R1 

The Xpert^®^ MTB/RIF diagnostic test for pulmonary and extrapulmonary tuberculosis in immunocompetent and immunocompromised patients: benefits and experiences over 2 years in different clinical contexts 

Dear Dr. Tomaz:

I'm pleased to inform you that your manuscript has been deemed suitable for publication in PLOS ONE. Congratulations! Your manuscript is now with our production department. 

Kind regards, 

on behalf of

Dr. Padmapriya P Banada 

Academic Editor

PLOS ONE